# Population Genetic Structure Reveals Two Lineages of *Amynthas triastriatus* (Oligochaeta: Megascolecidae) in China, with Notes on a New Subspecies of *Amynthas triastriatus*

**DOI:** 10.3390/ijerph17051538

**Published:** 2020-02-27

**Authors:** Yan Dong, Jibao Jiang, Zhu Yuan, Qi Zhao, Jiangping Qiu

**Affiliations:** 1School of Agriculture and Biology, Shanghai Jiao Tong University, Shanghai 200240, China; 2Key Laboratory of Urban Agriculture (South), Ministry of Agriculture, Shanghai 200240, China

**Keywords:** earthworms, megascolecidae, *Amynthas triastriatus*, phylogenetic, population genetic structure, molecular genetic

## Abstract

*Amynthas triastriatus* (Oligochaete: Megascolecidae) is a widely distributed endemic species in Southern China. To shed light on the population genetic diversity and to elucidate the population differentiation and dispersal of *A. triastriatus*, a population genetic structure study was undertaken based on samples from 35 locations collected from 2010 to 2016. Two exclusive lineages within *A. triastriatus*—lineage A and lineage B—were revealed. Lineage A was mainly distributed at high altitudes while lineage B was mainly distributed at low altitudes in Southeast China. The genetic diversity indices indicated that the populations of *A. triastriatus* had a strong genetic structure and distinct dispersal histories underlying the haplogroups observed in this study. Combined with morphological differences, these results indicated a new cryptic subspecies of *A. triastriatus*. Lineage A was almost degenerated to parthenogenesis and lineage B had a trend to parthenogenesis, which suggested that parthenogenesis could be an internal factor that influenced the differentiation and dispersal of *A. triastriatus*. The divergence time estimates showed that *A. triastriatus* originated around Guangxi and Guangdong provinces and generated into two main lineages 2.97 Ma (95%: 2.17–3.15 Ma) at the time of Quaternary glaciation (2.58 Ma), which suggested that the Quaternary glaciation may have been one of main factors that promoted the colonization of *A. triastriatus*.

## 1. Introduction

Understanding the evolutionary history of a species is a fundamental goal of evolutionary biology [1] and many approaches are based on population genetic structure analysis, both in fauna and flora [2,3]. Population genetic structure studies combined with geographical distribution, time of differentiation and ancient geographical events are commonly used to determine the differentiation and dispersal of populations and to analyze their influencing factors.

Earthworms are very important macro-invertebrate detritivores that tend to remain in the same areas during long periods of time and show low ability to cross mountains; for these reasons they are often an ideal model for phylogeographical study [4,5,6,7,8]. Earthworms play important roles in the agroecosystem as key organisms that influence soil structure formation, soil carbon dynamics and biogeochemical cycles [9,10,11], and hence failure to recognize accurate species boundaries within earthworms compromises many aspects of applied ecological, biodiversity, systematic, and evolutionary studies [12,13,14].

Their structural simplicity makes it difficult to distinguish earthworm species accurately using only morphological characteristics [15]. Molecular tools could be helpful to solve this problem, and analysis of population genetic structure and colonization history could reveal differentiation and dispersal of species and provide a solid base for taxonomy. Recent studies on this aspect for earthworms have expanded rapidly. Phylogenetic analysis using different substitution rates for the CO1 gene was carried out to illustrate the differentiation and diffusion of earthworms [6,7,16]. Parthenogenesis, geological events, climatic changes and human beings are the factors that exert significant influence on the differentiation and diffusion of earthworms [16,17,18,19,20]. Additionally, phylogenetic analysis constructed by single gene or different gene combinations have led to an-increasing discovery of cryptic earthworm species [6,12,15,21,22], which are taxonomically important to earthworms.

In recent research, mitochondrial COI, COII, NDI, 12S rRNA and 16S rRNA have been widely used to study the phylogenetic evaluation, mitochondrial diversity and population genetic diversity of earthworm using multivariate concatenation [18,21,23,24,25,26,27,28,29]. Complete mitochondrial genomes and genomic features of four pheretimoid earthworms (*Amynthas longisiphonus, Amynthas corticis, Amynthas gracilis* and *Metaphire californica*) revealed that mitochondrial ATP8 exhibited the highest evolutionary rate and the NADH gene was second to it [30]. Nuclear 18S rRNA has been more helpful within recognized families, rather than to elucidate deep annelid relationships [31,32]. The entire 28S rRNA appears suitable for within-genus and sometimes even within-species complexity if the sequences are analyzed with other genes [15,17,24,33]. The focus of this study was the intraspecies level using mitochondrial genes.

*Amynthas triastriatus* (Chen 1946) has two pairs of spermathecal pores in segments 7/8–8/9 and belongs to the *aeruginosus*-group. It is named after the triangle formed by three genital papillae in each male pore zone and is a widely distributed endemic species in Southern China [34]. There are many populations of *A. triastriatus* that are similar in morphology due to a tendency to parthenogenesis; for example, some populations have a thin and lustreless seminal chamber and prostate glands absent, and others have a plump and glossy seminal chamber and small prostate glands. As a result, their classification status and the genetic relationships are obscure.

A previous study based on only seven individuals indicated that the ancestor of *A. triastriatus* originated in the vicinity of Guangxi and Guangdong provinces then split into two lineages around 4.31 Ma (3.24–5.37 Ma) [34]—this was a little earlier than the Quaternary glaciation in China. Therefore, we assume that *A. triastriatus* presents two lineages, one displays parthenogenesis and the other is degenerate towards parthenogenesis. This raised general questions as whether there are cryptic species, whether the population genetic structure are useful to clear the classification status of *A. triastriatus* and whether genetic isolation occurred between the two lineages due to population expansion and reduction along with ancient geographical events. In the present study, a fragment of mitochondrial gene COI of 65 individuals collected from 35 locations and one *Aporrectodea* species, and 4691bp segments comprising mitochondrial genes COI, COII, ATP8, ND6, 12S rRNA, 16S rRNA and NDI from 35 locations and one *Aporrectodea* species were used to determine: (1) the population genetic diversity and the classification status of *A. triastriatus*; (2) the population differentiation and dispersal of *A. triastriatus*; (3) the time of divergence and colonization histories of *A. triastriatus*, as well the main factors impacting these.

## 2. Materials and Methods

### 2.1. Earthworm Sampling

A total of 232 individuals of *A. triastriatus* were collected by digging at 35 different locations (Figure 1, Table 1) from 17 provinces in China from 2010 to 2016. One individual of *Aporrectodea trapezoides* was collected as outgroup in Yunnan Province, China. Collected individuals were washed by clean water, anaesthetized in 10% ethanol solution, then fixed in anhydrous alcohol and preserved at 4 °C. The tail muscle issue (±0.02 g) was removed and cleaned carefully under a stereomicroscope for DNA extraction, and the rest of the individual was used for taxonomic identification.

### 2.2. DNA Extraction, Amplification and Sequencing

Total genomic DNA was extracted using the E.Z.N.A.^®^ Mollusc DNA kit (Omega Bio-Tek, Norcross, GA, USA) according to the manufacturer’s instructions. The quality of DNA samples was checked by electrophoresis using a 1% agarose gel and stored at −20 °C.

The mitochondrial genes were amplified using 50 μL of reaction mixture with 1 μL template DNA, 0.6 μL Trans Taq^®^ HIFI DNA Polymerase (Trans, Beijing, China), 4 μL dNTP, 5 μL HiFi Buffer, 2 μL primer and 35.4 μL ddH_2_O. The PCR of genes COI, COII, ATP8, 12S rRNA, 16S rRNA and NDI were carried out for 5 min at 94 °C followed by 32 cycles of 94 °C for 30 s, 50 °C for 30 s and 72 °C for 1 min, with an extension of 10 min at 72 °C. The PCR of gene ND6 was carried out for 5 min at 94 °C followed by 40 cycles of 94 °C for 30 s, 51.9 °C for 30 s and 72 °C for 1 min, with an extension of 10 min at 72 °C. The primers of mitochondrial COI, COII, 12S rRNA, 16S rRNA and NDI are listed in Table 2. The primers of genes ATP8 and ND6 were designed in this paper.

The PCR products were visualized on a 1% agarose gel and subsequently purified then sequenced by the Beijing Genomics Institute (Shanghai, China) using an ABI 3730 DNA analyzer. The sequences were compared with known earthworm sequences in GenBank using the BLAST search algorithm [35,36,37,38].

### 2.3. Sequence Alignment, Population Genetic Structure and Genetic Diversity, Time of Divergence and Colonization History

Population genetic structure and genetic diversity studies were conducted using single-gene datasets of COI with one *Aporrectodea* species as an outgroup. Time of divergence and colonization history were conducted using a combination of COI, COII, ATP8, ND6, 12S rRNA, 16S rRNA and NDI datasets with one *Aporrectodea* species as an outgroup. Sequences of COI and a combined dataset were aligned using the Clustal X 1.8 program [39].

Models for sequence evolution and corresponding parameters were estimated using jModeltest v.2.1.7 [40]. The best-fitting models were used in the phylogenetic estimation for the maximum likelihood (ML) and Bayesian phylogeny estimation (BI) analyses. jModeltest revealed TPM3uf+G to be the best-fitting model for the COI gene (679 bp) with a gamma shape parameter of 0.2360, and GTR+G to be the best-fitting model for 4691 bp segments comprising mitochondrial genes COI, COII, ATP8, ND6, 12S rRNA, 16S rRNA and NDI with a gamma shape parameter of 0.3950.

The phylogenetic trees were reconstructed using different methods based on single-gene COI datasets of 65 *A. triastriatus* individuals and one individual of *Aporrectodea trapezoides* as outgroup. ML analyses were performed using PhyML v.3.0 [41]; clade support was assessed using bootstrapping with 1000 pseudo replicates. BI was performed using MRBAYES v.3.2 [42]; the parameters were set to 10,000,000 with trees sampled every 1000 generations, discarding the first 10% as burn-in. Posterior probabilities and bootstrap support for each branch were calculated from the sampled trees.

Population structure was evaluated in Arlequin v.3.5.2 [43] via AMOVA with 10,000 permutations for statistical confidence. A haplotype network was constructed using the median joining (MJ) method Network v.4.6.1.6 [44] to infer the relationships among haplotypes and their geographical distribution. Nucleotide mismatch distribution analyses of lineages A and B were calculated with DnaSP 5.0, and neutrality tests were used to test population equilibrium.

Basic statistics of mtDNA diversity, including nucleotide and haplotype diversity, Tajima’s *D* [45] and Fu’s *Fs* [46] neutrality tests and nucleotide mismatch distribution analysis with and within lineages were calculated with DnaSP 5.0 [47]. Bayesian Skyline Plot (BSP) was constructed with BEAST 1.8.2 [48] and Tracer 1.7 [49] to infer the timing of the population events.

The Bayesian tree and the time of divergence of *A. triastriatus* were estimated using BEAST 1.8.2 [48] based on 4691 bp segments comprising mitochondrial genes COI, COII, ATP8, ND6, 12S rRNA, 16S rRNA and NDI. These were from 18 individuals from 18 locations of lineage A, 17 individuals from 17 locations of lineage B and one individual of *Aporrectodea trapezoides* as an outgroup. A previous study has estimated 2.1 My^−1^ as the average substitution rate for COI, COII, ATP8, ND6, NDI, 12S and 16S genes [50], and this value was applied in the present study. Since earthworm lack a fossil record, the age of the calibration clade between *A. triastriatus* and *Aporrectodea trapezoides* was set at a mean age of 212 Ma (95%: 200–230 Ma); this represented the minimum age of Lumbricoidea and Megascolecoidea in our tree and is considered to be when the breakup and drift of Gondwana occurred [50,51,52,53]. Five replicate runs were performed with 10,000,000 MCMC steps, and the first 10% served as burn-in. The output was visualized in Tracer v1.6 to determine whether convergence and suitable effective sample sizes were achieved for all parameters. The maximum clade credibility tree was accessed using Tree Annotator v1.8.2 [48]. The tree was visualized and edited using Fig Tree v2.1.4.

## 3. Results

### 3.1. Population Genetic Structure and Genetic Diversity

The posterior probabilities from the Bayesian analysis and the bootstrap values from the ML analysis were plotted on the ML tree (Figure 2). *A. triastriatus* fell into two exclusive lineages, A and B, and this result was strongly supported in both the BI tree (1.00) and ML tree (100%). Lineage A consisted of four clades (clades 1–4) and lineage B consisted of two clades (clades 5, 6). Clade 1 was mainly distributed in Guizhou, Hubei, Chongqing and Sichuan provinces (10 of 15 individuals), clades 2 and 3 were mainly distributed in Anhui and Fujian provinces (six of 11 individuals) and clade 4 was mainly distributed in Guangxi, Guangdong and Hunan provinces (six of eight individuals). Clade 5 was mainly distributed in East China, such as Zhejiang, Anhui province, east of Jiangxi and Fujian province (18 of 22 individuals) and clade 6 was mainly distributed in the west of Southern China, such as Sichuan, Hunan, Guangdong, Guangxi and Guizhou provinces (six of nine individuals).

The haplotype network was constructed using a 679 bp fragment of COI from 65 individuals from the 35 locations. It exhibited a certain level of population genetic structure indicating that *A. triastriatus* had two exclusive lineages, which was in concordance with the phylogenetic trees (Figure 3). The topological relationship of lineage A fell into four clades. Clade 1 included Hap_1-8, clade 2 included Hap_9-11, clade 3 included Hap_12, two individuals from Anhui province and one individual from Zhejiang province of Hap_13, and clade 4 included the remainder of Hap_13 as well as Hap_14. Lineage B fell into two clades. Clade 5 included 22 individuals of Hap_15 and clade 6 included the remainder of Hap_15 as well as Hap_16 and Hap_17.

All 65 individuals of the 35 locations yielded high-quality DNA and were successfully sequenced for the COI gene. Sequences were deposited in GenBank under accession numbers shown in Appendix A. The mean base composition of the fragment showed a strong bias of A+T (A: 35.03%; C: 18.96%; G: 16.02%; T: 29.99%), as commonly found in invertebrate mitochondrial genomes [54]. The network showed that there were 31 nucleotide changes between lineages A and B, one within lineage B and 1–2 within lineage A. The sequenced region contained 16 polymorphic sites defining 14 haplotypes of lineage A, two polymorphic sites defining three haplotypes of lineage B and 49 polymorphic sites defining 17 haplotypes of *A. triastriatus* (Table 3). Haplotype and nucleotide diversities were used as measures of genetic diversity between and within lineages. General haplotype diversity was 0.871 for lineage A but only 0.123 for lineage B, and was 0.758 for *A. triastriatus*. The nucleotide diversity was low in lineage A and extremely low in lineage B. The genetic distances were 0.5% for lineage A, 0.1% for lineage B and 6.3% for *A. triastriatus*. These results showed that lineages A and B had exclusive haplotypes and suggested a great genetic differentiation exists between lineages.

### 3.2. Time of Divergence and Colonization History

As shown in Figure 4, the root of the tree, corresponding to the diversification of *A. triastriatus* and *Aporrectodea trapezoides,* was estimated to occur about 215.28 Ma (95%: 200.41–229.89 Ma). *Amynthas triastriatus* became two main lineages about 2.97 Ma (95%: 2.17–3.15 Ma) (Figure 4, Table 4).

Hierarchical population structure in *A. triastriatus* was studied using AMOVA to check structure at the lineage level based on a fragment of the COI gene (Table 5). The percentage of variation among lineages was 93.89%, and within lineages was just 6.11%. The F*_ST_* value was 0.9389 (N_m_ = 0.02), indicating significant population genetic structure. The value of pairwise comparisons of Φ_ST_ among lineages was significantly high, suggesting a great genetic differentiation exists between lineages A and B.

Tajima’s *D* values were negative within lineages, but positive and significant between lineages, indicating that the COI gene was possibly not a neutral evolution. Fu’s *Fs* test is more sensitive to detect population expansion [46]; Fu’s *Fs* test values were negative and nonsignificant within lineages, indicating that lineages A and B had experienced demographic expansion. Fu’s *Fs* test values were positive and nonsignificant between lineages, indicating that *A. triastriatus* had experienced demographic expansion. The results of nucleotide mismatch distribution analysis based on fragments of COI, COII, ATP8, ND6, 12S rRNA, 16S rRNA and NDI genes showed the following: a transition between unimodal and bimodal distributions for COI and 12S, the same as expected for COII, 16S, ND1, ATP8 and ND6 of lineage A; unimodal distribution for COI and ND1, the same as expected for COII, 16S, ATP8 and ND6; and bimodal distribution for the 12S gene of lineage B. The results revealed that lineage A (Figure 5a) and lineage B (Figure 5b) were not steady-state lineages, suggesting an older expansion. BSP revealed an explicit demographic history for the populations of *A. triastriatus* (Figure 6): a flat population history during previous climate fluctuations of last glacial period (MIS 2-4), a sudden population decline during last glacial period, a rapid population expansion after last glacial period.

## 4. Discussion

### 4.1. Application of Molecular Techniques in Taxonomy and Differentiation of Earthworms

Earthworms are important soil animals, but their simple structure and burrowing nature makes their taxonomy challenging and has led to a gross underestimation of the true level of earthworm biodiversity [55]. DNA taxonomy and associated molecular tools might be the only way to reveal the true level of biodiversity [56]. In recent years there has been increasing research use of sequence data along with morphological characters to distinguish earthworm species and to erect cryptic species [6,57,58].

Through analysising the population differentiation and diffusion process, population genetic structure studies and phylogenetic studies could provide an important basis to illustrate the classification statue of species. In previous study, phylogenetic trees and high haplotypes diversity indicated two distinct clades of *Drawida japonica* Michaelsen, 1892 collected in Shandong and Liaodong peninsulas [17]. A high genetic diversity suggests the presence of five cryptic allopatric species collected from the central Iberian Peninsula [15]. Lineages of *A. triastriatus* had a strong genetic structure and distinct demographic histories underlying the haplogroups observed in this study. The phylogenetic trees showed that *A. triastriatus* had two exclusive lineages, a differentiation was also shown by AMOVA, high Φ_ST_ index and the high percentage of variation between lineages. Taken together, these results indicated a high degree of isolation and a great genetic differentiation exists between two lineages.

Taxonomically speaking, COI gene divergence of *Aporrectodea caliginosa*, *Lumbricus castaneus*, *Lumbricus terrestris* and *Satchellius mammalis* was below 4% and ranged between 0.0% and 2.0% within *Aporrectodea longa* [12,59]. The interspecific COI gene divergence values of invasive Asian earthworms in the northeast United States ranged from 15.84% to 24.03%, and intraspecific COI gene divergence values ranged from 0.01% to 0.4% [29]. In this study, the COI gene divergence between the two lineages of *A. triastriatus* was 6.3%, which is neither in the intraspecific or interspecific range. However, there were also certain morphological differences between the two lineages. Thus, it is proposed to erect a new subspecies of *A. triastriatus* (Chen 1946), named *Amynthas triastriatus usualis* subsp. nov.; the description for this proposed subspecies is included as the final part of this paper.

### 4.2. Parthenogenesis and the Demographic and Dispersal of A. triastriatus

Parthenogenesis is a unisexual reproduction pattern corresponding to bisexual reproduction that is present in a few animal and plant species. Generally, the loss of sexual reproduction has been thought a dead end in evolution, leading to early extinction [60,61]. But in contrast to this perception, parthenogenesis has been observed in many species of earthworm (Lumbricidae and Megascolecidae) and parthenogenesis in earthworms is often related to polyploidy or aneuploidy [62,63,64,65,66,67,68,69,70]. Muldal (1952) points out that parthenogenesis is important as it makes the retention of polyploidy possible, and also favors the spread of polyploid forms into new areas, since even a single parthenogenetic individual may establish a population. The association between parthenogenesis and high polyploidy in earthworms produces an unexpected level of heterozygosity, an advantageous condition that provides resistance to environmental stress [65,69].

Parthenogenesis often results in polymorphism in earthworms [67], with morphological variability mainly relating to the reduction of reproductive structures such as seminal vesicles, spermathecae, prostates, and an empty seminal chamber. [69]. In the case of *A. triastriatus,* lineage A with a thin and lustreless seminal chamber and no prostate gland observed was almost degenerated to parthenogenesis, while lineage B with a plump and glossy seminal chamber and small prostate glands had a tendency to parthenogenetic reproduction. Compared to their sexual relatives, parthenogenetics often occur at high latitudes, high altitudes, on islands or island-like habitats, in xeric environments or in disturbed habitats [71,72,73,74,75]. Lineage A was mainly distributed at high altitudes in Southwest China, whereas lineage B was mainly distributed at low altitudes in Southeast China. Unisexual lineages can be highly ecologically successful, with broad geographical ranges that overlap and/or are more extensive than those of the sexual ancestor [76]. For example, the locations of A4 and B5, A8 and B7, A18 and B14 overlapped, and the AMOVA F_ST_ values indicated that lineages A and B had a high degree of isolation and a significant genetic differentiation exists with lineage A having greater genetic diversity than lineage B. These findings supported those of Simon et al. (2003), that unisexual lineages with high genetic diversity can live in wider geographical ranges than their sexual ancestor.

Based on phylogenetic studies, parthenogenetic lineages usually occupy terminal nodes of phylogenetic trees; therefore, unisexual forms are usually assumed to arise from sexual congeners [76]. The phylogenetic trees produced from 321 species of Megascolecidae earthworms from China showed that all of these species fell into 14 groups, with *A. triastriatus* located at the terminal node of the terminal clade of the fourteenth group [52]. In the present study, lineage A was at the terminal node too. Additionally, the time of divergence for lineage A (0.48 Ma; 95%: 0.22–0.53 Ma) was later than for lineage B (0.72 Ma; 95%: 0.29–1.47 Ma), indicating that these unisexual forms did indeed arise from sexual congeners, which was in accordance with previous findings [28,53,77].

### 4.3. Differentiation and Colonization of A. triastriatus

The divergence time estimate of 215.28 Ma for outgroup *Aporrectodea trapezoides* (95%: 200.41–229.89 Ma) was not significantly different from previous estimates of the diversification between Lumbricidae and Megascolecidae at about 200–220 Ma [50,51,52,53]. Combining the results of population genetic structure and phylogenetic analysis, as well as paleogeographic events and paleoclimate, it was attempted to determine the differentiation and colonization routes of lineages A and B. In a previous study, the Megascolecidae family of China was separated into 14 groups after *pheretima* spread into China from the Indo-China Peninsula, then spread to the east or north [52]. Additionally, the split between *A. triastriatus* and GX201201-11A (an unknown species in the same cluster as *A. triastriatus* collected in Guangxi province, China) was estimated to have taken place 6.49 Ma (95%: 5.23–7.88 Ma) in Guangxi province [34].

It is proposed that *A. triastriatus* then formed into two main lineages (A and B) at around 2.97 Ma (95%: 2.17-3.15 Ma), which happened together with the Quaternary glaciation (2.58 Ma). The diversification of lineage A was divided into three stages.

The first stage (node 18 in Figure 4) occurred at the end of Zhonglianggan glaciation (MIS 12) and resulted from the Kunlun-Yellow River tectonic movement (1.2–0.6 Ma BP) that occurred at the transition from the Early to Middle Pleistocene [78,79]. Lineage A (Hap_14) dispersed to Guangdong province (Figure 7). The Heishiding Mountains in Guangdong province was a possible glacial refugium for *Liquidambar* species during the Quaternary glaciation [80], and the network analysis showed that hap_14 (GD56, collected in Heishiding mountains) was the hap that connected to lineage B. And BSP analysis shown that the Quaternary glaciation have influence on the diffusion of *A. triastriatus*, significantly several refuge should exist during this period. Therefore, the Heishiding Mountains is proposed as a potential refugium for lineage A.

The second stage (nodes 19, 20, 21 and 25) occurred during MIS 7, 8, 9 and 10 during the Penultimate Glacial Period and before MIS 6, during which glaciation remained widespread with the maximum temperature decrease experienced in Western China and alpine areas [78,81,82,83,84]. Lineage A dispersed north to the mountainous terrain in the Sichuan, Chongqing and Guizhou provinces. Guizhou province, with its comparably stable ecological conditions during environmental fluctuations, is a refugium for *Ginkgo biloba* [85,86], and lineage A populations from Guizhou province contained 4 haplotypes with high levels of genetic diversity. Therefore, the Sichuan, Chongqing and Guizhou provinces are proposed as potential refugia for lineage A.

The third stage occurred during the last glacial period (MIS 2-4), in which the Gonghe Movement (~0.15 Ma BP) occurred between the Middle and Late Pleistocene to raise the Tibetan Plateau to its present height, High-Asia experienced several temperature plunges, and glaciation existed in most of the mountains in western China [87,88,89,90]. Lineage A dispersed to the east to Fujian province, followed by dispersal to Guangxi province and north to Hunan and Jiangxi provinces. Humid and suitable climate conditions in South China promoted its colonization.

Unlike lineage A, lineage B mainly dispersed to the east. The diversification of lineage B was also divided into three stages. The first stage (nodes 2, 3, 4 and 6 in Figure 4) occurred during the Wangkun Glaciation (MIS 16-20) in which Kunlun-Yellow River tectonic movement not only strongly uplifted the Tibetan Plateau but also the surrounding mountains. Coverage of the Tibetan Plateau by ice sheet peaked during MIS 16 at 18 times the present glacial coverage [91,92]; the mid-Pleistocene Revolution (~0.9 Ma), the most significant transition boundary, led to a significant drop in global temperature and extension of the ice sheet [92]. In this period, lineage B dispersed to Guangdong, Fujian and Zhejiang provinces, far from the Tibetan Plateau. Guangdong (Heishiding), Fujian (Wuyi Mountain) and Zhejiang (Tianmu Mountain) were proposed to be refugia of flora and fauna [86,93]. A lineage B was mainly distributed in Zhejiang province and the surrounding area, it is suggested that this glaciation period blocked the colonization of additional locations by lineage B, resulting in refugia in Zhejiang province.

The second stage (nodes 11 and 13) occurred during MIS 6 and 8, two glacial periods of the Penultimate Glaciation, with lineage B still dispersed in Zhejiang province. The third stage, as for lineage A, occurred during the last glacial period (MIS 2-4). One route of lineage B dispersal was to Anhui and Jiangxi provinces and another route was north to Guizhou and Guangxi provinces. Geological and climatic changes during this period had a profound effect on the colonization of lineage B, and it is also possible that the distant dispersal of B12 was affected by other factors such as birds or human beings.

The population structure of *Drawida japonica* Michaelsen, 1892 were analysed using 79 samples obtained from the Shandong and Liaodong peninsulas. Results show that the intraspecific genetic diversity of *D. japonica* has been influenced by geographic isolation [17]. The results of population genetic structure analysis of hormogastrid earthworms (*Hormogaster elisae* Álvarez 1977) collected from the central Iberian Peninsula and some invasive species (*Amynthas corticis* (Kinberg, 1867), *Amynthas gracilis* (Kinberg, 1867)) collected from a volcanic island show that environmental factors may have some influence on earthworms’ genetic evolution [15,94]. The quaternary glaciation resulted in repeated drastic environmental changes that profoundly shaped the current distributions and genetic structure of many plants and animal species in temperate zones of the Northern Hemisphere [95,96,97]. As the youngest major glacial period, Quaternary glaciation ended in a recessional event extending from 0.015 to 0.01 Ma at mid-latitudes [98,99]. The present-day geographical distribution pattern of *A. triastriatus* formed about 0.01 Ma, with its dispersal process totally covered by the Quaternary glaciation period. This period had a major impact on climate and vegetation in Southeast Asia through the interaction of temperature, rainfall and topography [100,101]. We suggest that the Quaternary glaciation period promoted the differentiation and colonization of *A. triastriatus*. *Drawida japonica*, *H. elisae*, *A. corticis*, *A. gracilis* and *A. triastriatus* are wildspread species, ancient geography event, climate changes, geographic isolation (island, sea, river) and environment factors should play important role in the differentiation and diffusion of wildspread earthworm species.

The phylogenetic evaluation of the genus *Amynthas* was carried out using 77 *Amynthas* species from South China. It has been determined that at least one branch of *Amynthas* spread to the southeast of China, and another spread to the southwest of China [28]. In the present study, the spread of *A. triastriatus* was consistent with the research of Sun et al., with lineage A widely distributed in the southwest of China, and lineage B widely distributed in the southeast of China. Overall, the colonization routes of *A. triastriatus* were from south to north, which is in accord with the known dispersal direction of earthworms in China [52,102].

## 5. Conclusions

*Amynthas triastriatus* (Oligochaete: Megascolecidae) of the *aeruginosus*-group is a widely distributed endemic earthworm species in Southern China. Population genetic structure exhibited a very high degree of isolation and two exclusive lineages based on existing specimen and gene data. Lineage A was almost degenerated to parthenogenesis, lineage B had a tendency to parthenogenesis. There were distinguish genetic differentiations exist between lineages and lineage A had high haplotype diversity and strong genetic structure, additionally, lineage A mainly distributied at high alitude areas. These results indicated a new cryptic species of *A. triastriatus*; combined with morphological differences, a new subspecies of *A. triastriatus* is proposed. And the Quaternary glaciation may be one of main factors that promoted the colonization of *A. triastriatus*. In future, there may be more studies of differentiation and dispersal of *A. triastriatus* including more populations based on this study, and the population genetic structure analysis should play an important role in the earthworm taxonomy.

## 6. Description of a New Subspecies of Amynthas Triastriatus

Based on the above results, to unravel the classification puzzle of *Amynthas triastriatus* it is proposed that a new subspecies be established that encompasses the populations of lineage B.

**Material. Holotype:** 1 clitellate (C-FJ201111-04A): China, Fujian Province, Meihua Mountain Nature Reserve (25.31278° N, 116.89306° E), 1207 m asl, black soil beside road, 13th July, 2011, J. B. Jiang, J. Sun, X. D. Lei and H. W. Feng coll. **Paratypes:** 85 clitellates: 2 clitellates (C-FJ201111-04B) same date as for holotype. 4 clitellates (C-FJ201004-01): China, Guangxi Province, Youxi Country (26.11244° N, 118.42387° E), 952 m asl, black soil beside road, 24th Aug, 2010, J. B. Jiang coll. 1 clitellate (C-GD201115-04): China, Guangdong Province, Lianping county (23.46417° N, 114.91444° E), 410 m asl, brown soil under eucalyptus trees, 04th Aug, 2011, J. B. Jiang coll. 12 clitellates (C-GX201319-06): China, Guangxi Province, Jiuwan Mountain Nature Reserve (25.20595° N, 108.67555° E), 1226 m asl, yellow soil, 18th May, 2013, J. P. Qiu, Y, Hong, J. B. Jiang, L. L. Zhang and Y. Dong coll. 4 clitellates (C-GZ201609-02): China, Guizhou Province, Fanjing Mountain National Reserve (27.52480° N, 108.59300° E), 1316 m asl, yellow soil under bryophyte beside road, 21th May, 2013, J. P. Qiu, J. B. Jiang, L. L. Zhang and Y. Dong coll. 2 clitellates (C-HU201309-02): China, Hunan Province, Bamian Mountain Nature Reserve (27.25680° N, 112.72433° E), 1034 m asl, black soil under willow trees, 21th Jun, 2013, J. P. Qiu, J. B. Jiang, L. L. Zhang and Y. Dong coll. 2 clitellates (C-JX201302-04): China, Jiangxi Province, Longhu mountain (28.03037° N, 117.09680° E), 185 m asl, black soil beside road, 15th Jun, 2013, J. B. Jiang, L. L. Zhang, Y. Dong and S. Lei coll. 19 clitellates (C-JX201311-01): China, Jiangxi Province, Jinggang Mountain National Reserve (26.55213° N, 114.10773° E), 1300 m asl, black soil under bamboo forest beside road, 17th Jun, 2013, J. B. Jiang, L. L. Zhang, Y. Dong and S. Lei coll. 4 clitellates (C-SC201004-06): China, Sichuan Province, Gongga Mountain National Reserve (29.65556° N, 102.11667° E), 1558 m asl, brown soil beside river, 06th Aug, 2010, J. P. Qiu, J. B. Jiang coll. 1 clitellate (C-ZJ201101-02): China, Zhejiang Province, Tianmu Mountain National Reserve (30.32886° N, 102.11667° E), 1052 m asl, black soil beside road, 02nd Nov, 2011, J. B. Jiang, J. Sun, L. L. Zhang coll. 1 clitellate (C-ZJ201511-01): China, Zhejiang Province, Jinhua City (28.83361° N, 119.24639° E), 160 m asl, black soil beside road, 08th July, 2015, Y. Dong, M. S. Chen, X. Gao and C. Peng coll. 4 clitellates (C-ZJ201515-03): China, Zhejiang Province, Yunhe City (28.10444° N, 119.63472° E), 290 m asl, yellow soil beside road, 09th July, 2015, Y. Dong, M. S. Chen, X. Gao and C. Peng coll. 1 clitellate (C-ZJ201517-02): China, Zhejiang Province, Wenzhou City (27.84556° N, 119.77306° E), 710 m asl, brown soil beside road, 09th July, 2015, Y. Dong, M. S. Chen, X. Gao and C. Peng coll. 6 clitellates (C-ZJ201518-01): China, Zhejiang Province, Yandang mountain (28.37639° N, 121.10972° E), 120 m asl, brown soil beside road, 10th July, 2015, Y. Dong, M. S. Chen, X. Gao and C. Peng coll. 10 clitellates (C-AH201601-01): China, Anhui Province, Huangshan City (29.95601° N, 118.09112° E), 325 m asl, brown soil, 08th May, 2016, J. B. Jiang, J. Sun, Y. Dong and Y. Zheng coll. 2 clitellates (C-AH201609-02): China, Anhui Province, Huangshan City (30.31276° N, 118.02561° E), 120 m asl, yellow soil, 09th May, 2016, J. B. Jiang, J. Sun, Y. Dong and Y. Zheng coll. 10 clitellates (C-AH201610-01): China, Anhui Province, Huangshan City (30.35924° N, 117.93609° E), 146 m asl, brown soil, 09th May, 2016, J. B. Jiang, J. Sun, Y. Dong and Y. Zheng coll.

**Diagnosis.** Dimensions 120–150 mm by 4.9–6.8 mm at clitellum, segments 108–111. First dorsal pore in 11/12. Setae numbering 20–26 at III, 24–30 at V, 30–34 at VIII, 40–48 at XX and 55–60 at XXV; 12–16 between male pores; seta numbering between spermathecal pores 10–14 at VIII. Two pairs of spermathecal pores in 7/8–8/9, 1/3 circumference ventrally apart from each other. One pair of male pores in XVIII, 1/3 circumference apart ventrally, each on the top of a raised, round porophore surrounded by 3–5 circular ridges. Two presetal and two postsetal collapse-topped genital papillae present on XVIII. Ampulla oval-shaped, stout duct as long as 1/5 ampulla. Diverticulum as long as 1/3 main pouch, terminal 1/2 dilated into an oval-shaped glossy seminal chamber. Prostate glands small or only one side observed.

**Description. External characters:** Brown pigment on dorsum, no pigment on ventrum. Dimensions 125 mm by 5.1 mm at clitellum, segments 111. Prostomium 1/2 epilobous. First dorsal pore in 11/12. Setae numbering 22 at III, 26 at V, 32 at VIII, 44 at XX and 58 at XXV; 14 between male pores; 14 at VIII between spermathecal pores. Setal formula: AA = 1.2 − 1.4AB, ZZ = 1.6 − 2.0ZY. Clitellum annular, taupe on dorsal, in XIV–XVI, setae not visible external. Two pairs of spermathecal pores in 7/8–8/9, 1/3 circumference apart ventrally. Two pairs of postsetal genital papillae present on VII, two pairs presetal and two pairs postsetal on VIII, three pairs presetal on IX, 1/3 circumference ventrally apart from each other (paratypes: numbers of genital papillae variable from 14–18). One pair of male pores in XVIII, 1/3 circumference apart ventrally, each on the top of a raised, round porophore surrounded by 3–5 circular ridges, two presetal and two postsetal genital papillae on median side. Two big collapse-topped genital papillae present on XVIII, 1/4 circumference ventrally apart from each other. (Figure 8A). Single female pore in XIV, milky.

**Internal characters.** Septa 5/6–7/8, thick and muscular, 10/11–12/13 slightly thickened, 8/9–9/10 absent. Gizzard bucket-shaped, in IX–X. Intestine enlarged distinctly from XV. One pair of intestinal caeca in XXVII, brown, simple, dorsal margin smooth, 17 short pointed saccules in ventral margin, extending anteriorly to XXIII. Four pairs of esophageal hearts in X–XIII. Ovaries in XIII. Two pairs of spermathecae in VIII–IX, about 1.7–2.0 mm long. Ampulla oval-shaped, stout duct as long as 1/5 ampulla. Diverticulum as long as 1/3 main pouch, terminal 1/2 dilated into an oval-shaped glossy seminal chamber. (Figure 8B). Holandric: two pairs of testis sacs, in X–XI undeveloped. Two pairs of seminal vesicles in XI and XII, the second pair more developed than the first pair. Holotype with one pair of thick, small, nubby lobate prostate glands and S-curved duct. Paratypes with one pair of prostate glands or only one side observed and other side S-curved duct (Figure 8C).

**Ethymology.** The species is named after the existence of prostate glands.

**Remarks.** There are certain differences between the new subspecies and *A*. *triastriatus*. For instance, the new subspecies has small prostate glands that are absent in *A*. *triastriatus*, and the seminal chamber of the new subspecies is plump and glossy while thin and lustreless in *A*. *triastriatus*. Additionally, the first dorsal pore of the new subspecies is located at 11/12, but at 10/11 in *A*. *triastriatus*. Furthermore, the new subspecies has more papillae in both the spermathecal and male pore regions than *A*. *triastriatus*, and more finger sacs in the ventral intestinal caeca than *A*. *triastriatus*.

## Figures and Tables

**Figure 1 ijerph-17-01538-f001:**
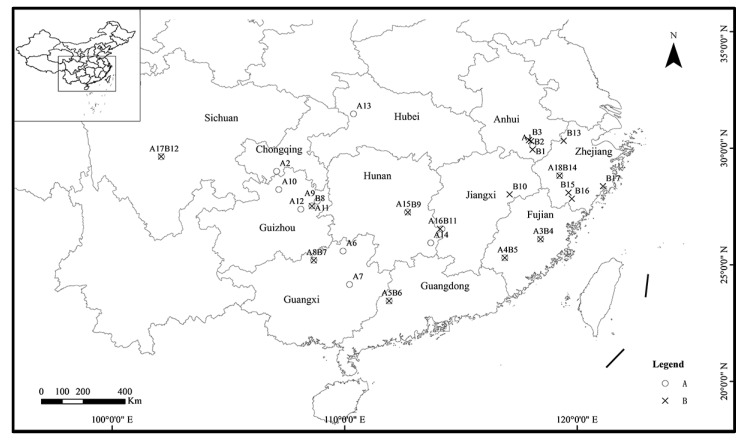
The 35 sampling locations in China. The circles represent 18 locations of lineage A. The crosses represent 17 locations of lineage B. Location details are shown in Table 1.

**Figure 2 ijerph-17-01538-f002:**
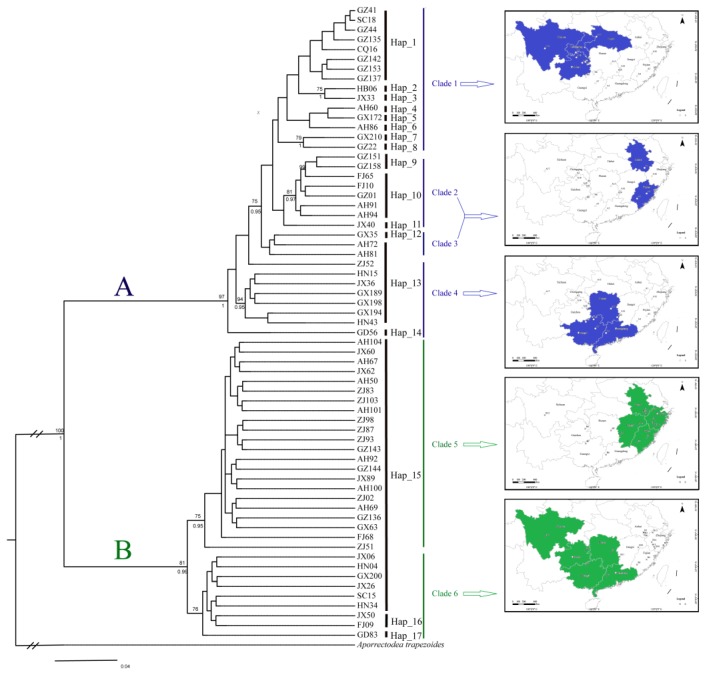
Phylogenetic trees reconstructed using different methods (ML and BI) based on the COI gene. Branch lengths are proportional to the amount of change along the branches. Bootstrap proportions (if ≥70%) and Bayesian posterior probabilities (if ≥0.95) are shown above and below the branches, respectively. *Aporrectodea trapezoides* was used as the outgroup taxa.

**Figure 3 ijerph-17-01538-f003:**
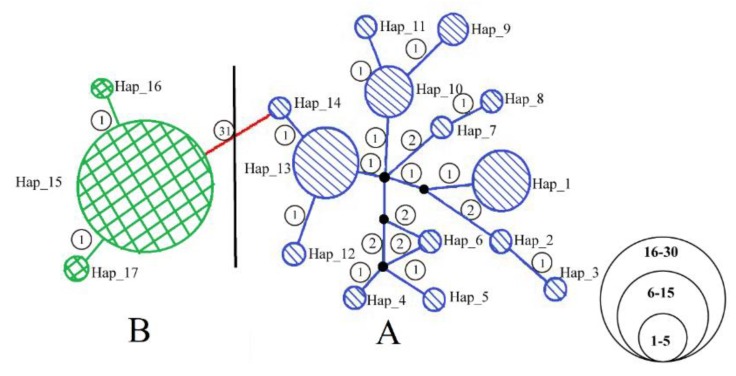
Unrooted network of 17 haplotypes in 65 A. triastriatus individuals from 35 locations (lineage A: blue; lineage B: green) based on 679 bp of a fragment of the mitochondrial COI gene. Each circle in the haplotype network corresponds to one haplotype, and its size is proportional to its frequency among samples. The numbers along branches represent the number of mutations between haplotypes. Dark dots are median vectors that represent intermediate haplotypes that were not sampled or are extinct.

**Figure 4 ijerph-17-01538-f004:**
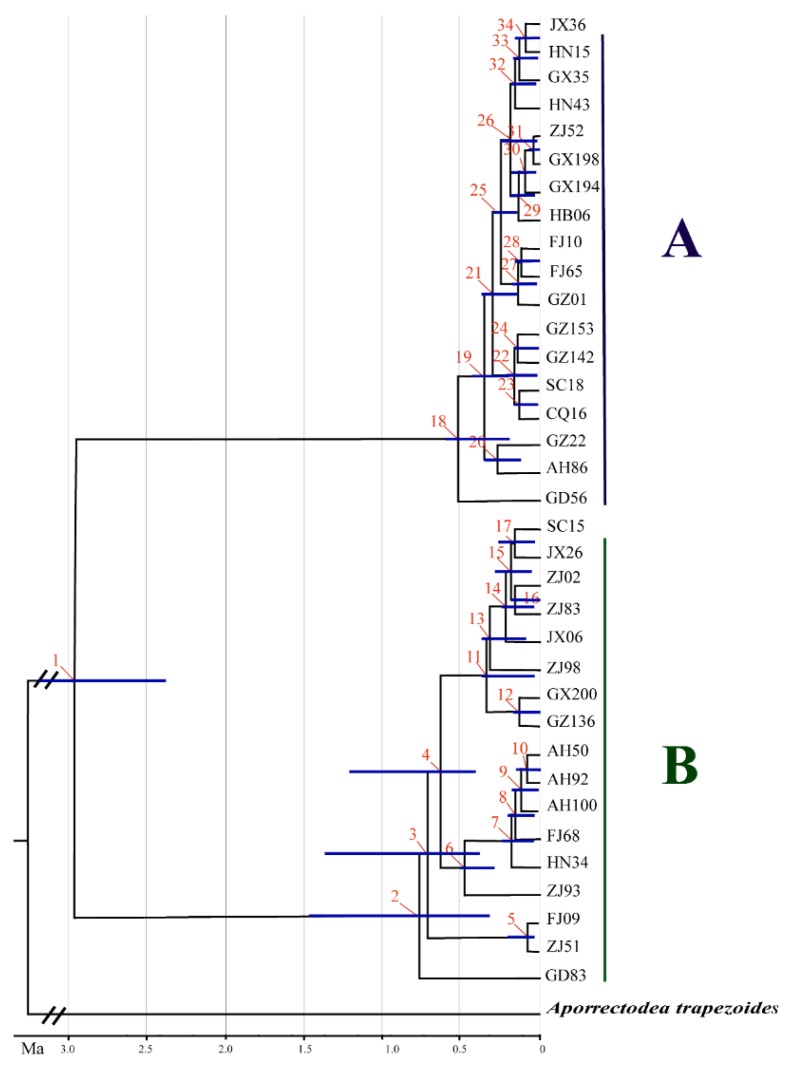
Divergence time estimate for *A. triastriatus* using a combined COI, COII, ATP8, ND6, 12S rRNA, 16S rRNA and NDI database based on a relaxed molecular clock (uncorrected lognormal). Details of numbers in each node are shown in Table 4.

**Figure 5 ijerph-17-01538-f005:**
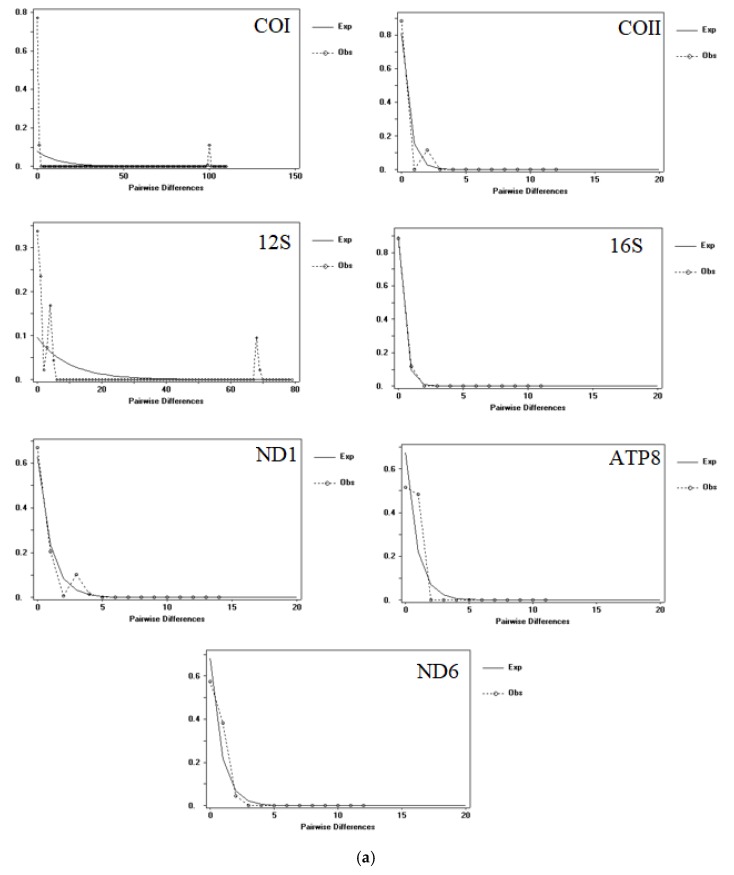
Graphs of the mismatch distribution of lineage A (a) and lineage B (b) based on mitochondrial genes COI, COII, ATP8, ND6, 12S rRNA, 16S rRNA and NDI. The abscissa represents the number of pairwise distances and the ordinate represents the number of observations. The lines represent the expected (smoothed) and observed (dashed) frequency of pairwise differences under the sudden lineage expansion model.

**Figure 6 ijerph-17-01538-f006:**
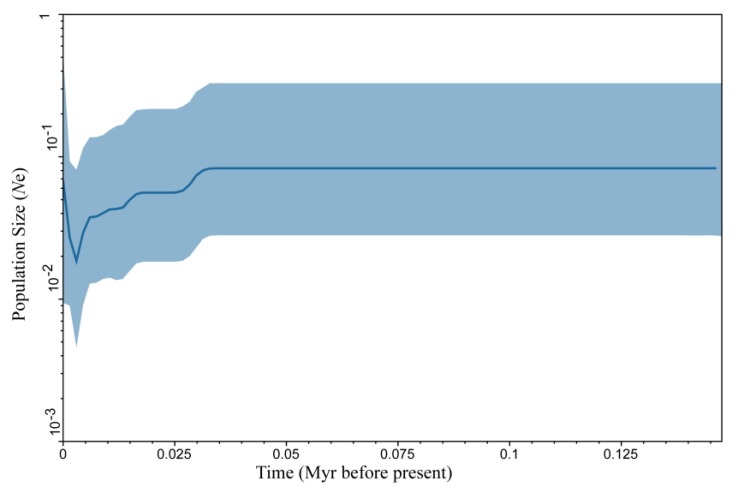
Bayesian skyline plots of population dynamic for populations of A. *triastriatus*. The shade areas in light blue are within the 95% highest posterior density interval. The dotted vertical lines indicate important population-size changes of *A*. *triastriatus*.

**Figure 7 ijerph-17-01538-f007:**
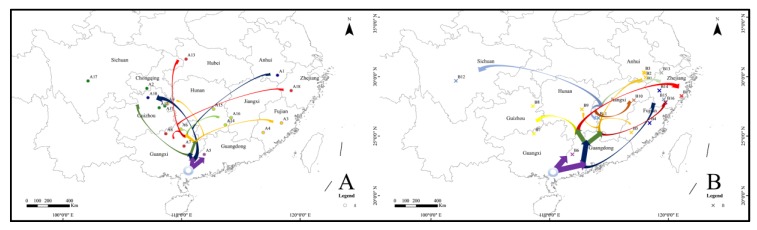
Dispersal routes of lineage A (**left**) and lineage B (**right**). Reconstructions are based on the network and genetic analysis. The arrows represent possible spread directions but not actual dispersal routes. Different colored arrows represent different branches in Figure 4.

**Figure 8 ijerph-17-01538-f008:**
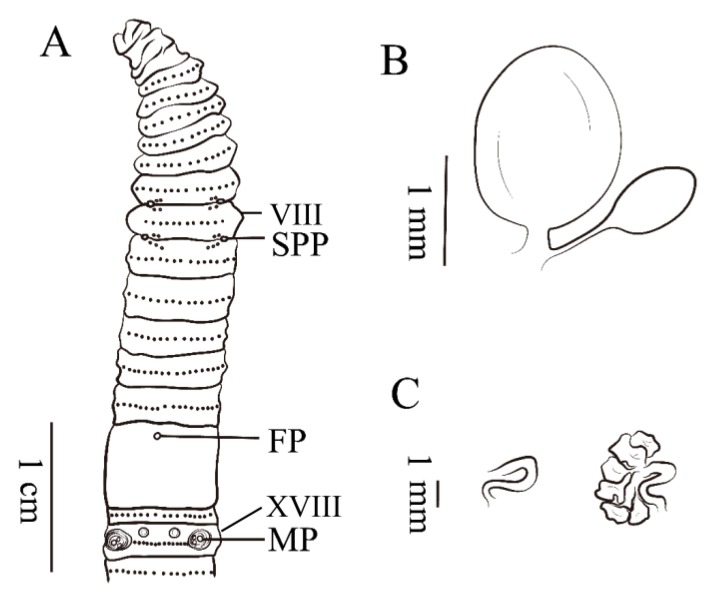
(**A**) Ventral view showing spermathecal pores, female pores and male pores of *A. triastriatus usualis*; (**B**) spermathecae of *A. triastriatus usualis*; (**C**) prostate glands of *A. triastriatus usualis*.

**Table 1 ijerph-17-01538-t001:** Collection data and information on *A. triastriatus* sequenced in this study using the COI gene. The code in bold formatting was sequenced using 7 genes.

Taxon	Label	Location Code	Collection Location	GPS Coordinates	Collection Date	Elevation (m)	Gene Code	Number of Individuals
A	C-AH201608-01	1	Anhui	30.12324° N, 118.09639° E	05/2016	93	**AH86** AH60 AH72 AH81 AH91 AH94	19
	C-CQ201304-02	2	Chongqing	29.01467° N, 107.08238° E	06/2013	1162	**CQ16**	2
	C-FJ201106-01	3	Fujian	26.10627° N, 118.42997° E	07/2011	342	**FJ10**	1
	C-FJ201111-01	4	Fujian	25.31278° N, 116.89306° E	07/2011	1207	**FJ65**	18
	C-GD201109-03	5	Guangdong	23.46417° N, 111.91444° E	10/2011	184	**GD56**	4
	C-GX201210-05	6	Guangxi	25.59723° N, 109.93519° E	06/2012	1536	**GX35**	13
	C-GX201318-05	7	Guangxi	24.15622° N, 110.21279° E	05/2013	1198	**GX194** GX172 GX189	20
	C-GX201319-04	8	Guangxi	25.20595° N, 108.67555° E	05/2013	1226	**GX198** GX210	2
	C-GZ201101-01	9	Guizhou	27.73722° N, 108.81611° E	10/2011	370	**GZ01** GZ41 GZ44	27
	C-GZ201103-02	10	Guizhou	28.23472° N, 107.17444° E	10/2011	1534	**GZ22**	4
	C-GZ201610-01	11	Guizhou	27.53520° N, 108.59300° E	10/2016	604	**GZ142** GZ135 GZ137	14
	C-GZ201611-05	12	Guizhou	27.39100° N, 108.11610° E	10/2016	890	**GZ153** GZ151 GZ158	3
	C-HB201101-06	13	Hubei	31.47611° N, 110.38055° E	09/2011	1399	**HB06**	1
	C-HU201304-01	14	Hunan	25.94700° N, 113.70489° E	06/2013	982	**HN15**	4
	C-HU201311-04	15	Hunan	27.27608° N, 112.70904° E	06/2013	500	**HN43**	1
	C-JX201313-02	16	Jiangxi	26.54490° N, 114.18954° E	06/2013	571	**JX36** JX33 JX40	10
	C-SC201006-02	17	Sichuan	29.64400° N, 102.12433° E	08/2010	1625	**SC18**	2
	C-ZJ201511-02	18	Zhejiang	28.83361° N, 119.24639° E	07/2015	160	**ZJ52**	1
B	C-AH201601-01	1	Anhui	29.95601° N, 118.09112° E	05/2016	325	**AH50** AH67 AH69	10
	C-AH201609-02	2	Anhui	30.31267° N, 118.02561° E	05/2016	120	**AH92**	2
	C-AH201610-01	3	Anhui	30.35924° N, 117.93609° E	05/2016	146	**AH100** AH101 AH104	10
	C-FJ201004-01	4	Fujian	26.11244° N, 118.42387° E	08/2010	952	**FJ09**	4
	C-FJ201111-04	5	Fujian	25.31278° N, 116.89306° E	07/2011	1207	**FJ68**	3
	C-GD201115-04	6	Guangdong	23.46417° N, 111.91444° E	08/2011	410	**GD83**	1
	C-GX201319-06	7	Guangxi	25.20595° N, 108.67555° E	05/2013	1226	**GX200** GX63	12
	C-GZ201609-02	8	Guizhou	27.52480° N, 108.59300° E	10/2016	731	**GZ136** GZ143 GZ144	4
	C-HU201309-02	9	Hunan	27.25680° N, 112.72433° E	06/2013	172	**HN34** HN04	2
	C-JX201302-04	10	Jiangxi	28.03037° N, 117.09680° E	06/2013	185	**JX06**	2
	C-JX201311-01	11	Jiangxi	26.55213° N, 114.10773° E	06/2013	1300	**JX26** JX50 JX60 JX62 JX89	19
	C-SC201004-06	12	Sichuan	29.65556° N, 102.11667° E	08/2010	1558	**SC15**	4
	C-ZJ201101-02	13	Zhejiang	30.32886° N, 119.42599° E	11/2011	1052	**ZJ02**	1
	C-ZJ201511-01	14	Zhejiang	28.83361° N, 119.24639° E	07/2015	160	**ZJ51**	1
	C-ZJ201515-03	15	Zhejiang	28.10444° N, 119.63472° E	07/2015	290	**ZJ83** ZJ87	4
	C-ZJ201517-02	16	Zhejiang	27.84556° N, 119.77306° E	07/2015	710	**ZJ93**	1
	C-ZJ201518-01	17	Zhejiang	28.37639° N, 121.10972° E	07/2015	120	**ZJ98** ZJ103	6

**Table 2 ijerph-17-01538-t002:** Primers used for PCR and sequencing.

Gene	Primer	Sequence	Source
CO1	LCO1490	5′-GGTCAACAAATCATAAAGATATTGG-3′	[36]
	HCO2198	5′-TAAACTTCAGGGTGACCAAAAAATCA-3	[36]
	CO1-E	5′-TATACTTCTGGGTGTCCGAAGAATCA-3′	[37]
CO2	rRNA-Asn-CO2-tRNA-Asp: LumbF1	5′-GGCACCTATTTGTTAATTAGG-3′	[33]
	rRNA-Asn-CO2-tRNA-Asp: LumbR2	5′-GTGAGGCATAGAAATACACC-3′	[33]
ATP8	ATP8-F	5′-GYTTAGTTRCCAMCYGGTGTATTTC-3′	This study
	ATP8-R	5′-CTTYYTACTTGGAAGGTARRTGTAC-3′	This study
ND6	ND6-F	5′-TGTATGGYGCACACRGGYHTTTGAA-3′	This study
	ND6-R	5′-RATDGCTGGRGTNGGYTTAAACATA-3′	This study
12S rRNA	12S-tRNA-Val-16S-LumbF1	5′-CTTAAAGATTTTGGCGGTGTC-3′	[33]
	12S-tRNA-Val-16S-LumbR1	5′-CCTTTGCACGGTTAGGATAC-3′	[33]
16S rRNA	16Sar	5′-CGCCTGTTTATCAAAAACAT-3′	[38]
	16Sbr	5′-CCGGTCTGAACTCAGATCACGT-3′	[38]
ND1	rRNA-Leu-ND1-LumbF2	5′-GAATAGTGCCACAGGTTTAAAC-3′	[33]
	rRNA-Leu-ND1-LumbR1b	5′-TTAACGTCATCAGAGTTATC-3′	[33]

**Table 3 ijerph-17-01538-t003:** Summary of genetic diversity measures and neutrality tests between and within lineages based on the COI gene, and the mean number of pairwise distances based on a fragment of the COI gene and on a combination of seven genes.

Taxon	Number of Individuals	Number Haplotypes	Polymorphic Sites	Haplotype Diversity	Nucleotide Diversity	Tajima’s D	Fu’s FS	Mean Number of Pairwise Distance	
								COI 7	Gene
A (within lineage)	34	14	16	0.866	0.0052	−1.028	−4.637	0.5%	0.6%
B (within lineage)	31	3	2	0.127	0.0002	−1.506	−2.397	0.1%	0.3%
Total (with lineage)	65	17	49	0.769	0.0309	2.303 *	7.525	6.3%	6.7%

* *p* < 0.05.

**Table 4 ijerph-17-01538-t004:** Divergence times of *A. triastriatus*, estimated by the Bayesian method.

Node	Estimated Age in Millions of Years Ago (Ma)		
	Age	Lower 95% Confidence Limit	Upper 95% Confidence Limit
1	2.97	2.17	3.15
2	0.72	0.29	1.47
3	0.69	0.27	1.42
4	0.60	0.26	1.35
5	0.08	0.01	0.13
6	0.42	0.26	0.41
7	0.09	0.04	0.14
8	0.07	0.03	0.11
9	0.03	0.01	0.06
10	0.02	0.00	0.04
11	0.26	0.03	0.31
12	0.02	0.00	0.04
13	0.15	0.02	0.19
14	0.05	0.01	0.05
15	0.03	0.01	0.04
16	0.02	0.00	0.03
17	0.02	0.01	0.04
18	0.48	0.22	0.53
19	0.33	0.19	0.39
20	0.24	0.17	0.28
21	0.27	0.15	0.35
22	0.04	0.01	0.06
23	0.02	0.00	0.04
24	0.01	0.00	0.03
25	0.23	0.14	0.25
26	0.09	0.04	0.11
27	0.04	0.01	0.06
28	0.02	0.00	0.04
29	0.06	0.03	0.09
30	0.04	0.01	0.07
31	0.02	0.00	0.03
32	0.05	0.02	0.07
33	0.03	0.01	0.05
34	0.01	0.00	0.03

**Table 5 ijerph-17-01538-t005:** ANOVA results between and within lineages based on COI a gene fragment. d.f. = degrees of freedom, Φ_ST_ = ΦStatistic, FST = genetic differences among population.

Source of Variation	d.f.	Sum of Squares	Variance Components	Percentage of Variation	Φ_ST_
Between lineages	1	629.617	19.37529 Va	93.89	0.9396
Within lineages	63	79.445	1.26103 Vb	6.11	(*p* < 0.0001)
Total	64	709.062	20.63632		
F_ST_	0.9389

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
