# Peer review of "Population Genetic Structure Reveals Two Lineages of *Amynthas triastriatus* (Oligochaeta: Megascolecidae) in China, with Notes on a New Subspecies of *Amynthas triastriatus"

_ijerph, 2020, doi:10.3390/ijerph17051538_

Round 1

Reviewer 1 Report

The manuscript “Population genetic structure reveals two lineages of Amysthas triastriatus (Oligachaeta: Megascolecidae) in China, with notes on a new subspecies of Amynthas triastriatus” investigated the pattern of genetic diversity distribution and population structure in A. triastriatus, an endemic earthworm in China. Dong et al. used six mitochondrial genes fragments to study the phylogenetic relationships, the presence of putative cryptic species, and the history of the expansion of species of A. triastriatus in the region. The authors used an adequate number of samples (~232) for this type of study collecting samples at 35 different locations. The analyses used are standard for this kind of work and were well executed. The study found evidence for the presence of two lineages phylogeographically separated that have diverged around ~2 Mya ago. Lineage A is distributed in high altitudes whereas lineage B occupies low altitudes. The authors have also found pieces of evidence that the lineage formation might also be associated with the evolution of parthenogenesis in this species. Lastly, due to the limited capacity of this species to disperse, most events of differentiation could be correlated with significant climate events such as glaciations and changes in topography.

The findings presented by Dong et al. are relevant and worth to be shared with the scientific community. I have only small comments and suggestions for the authors that might help them to improve the paper.

Minor corrections and suggestions

The maps presented are difficult to read. Perhaps the low resolution or the small font size makes it very difficult to read its content. Border and margins are also too thin. Figure 2, for instance, is very informative and the colored maps make a very compelling argument for the allopatric differentiation of linages and sub-lineage; however, the maps have insufficient resolution and are mostly faded. I would suggest that those maps be replotted with a thicker margin and better resolution.   

Ln 165-166. “The haplotype network was constructed using a 679 bp fragment of COI from 65 individuals of 35 populations.” I suggest changing the word “populations” for a more adequate term such as “collection sites” or “locations.” One of the main objectives of this work is to determine the number of gene pools (i.e., real populations), which the authors later found to be two and not 35.

Ln 174-175. “All 65 individuals of 35 population yielded high-quality (…).” Same as above.

Ln 212-213. “The value of pairwise comparisons of φST among lineages was significantly high, suggesting a lack of gene flow between lineages A and B.”  As mitochondrial markers do not recombine and therefore do not allow us to conclude anything about lineage hybridization of introgression, the authors should be cautious using expressions such as the above. The data indicate strong geographical isolation and genetic structure, which means that drift is more important than gene flow. However, are the lineages isolated because of the parthenogenetic reproduction (i.e.,  the gene pool is inaccessible to one another)? Can they hybridize at some point in the geographical range?  

The statements in Figure 5.1 and 6.1 captions (“observed frequency of pairwise difference under the sudden lineage expansion model”) contradicts the statement made in the line 226-227 (“(…) were not steady-state lineages, suggesting an old expansion”). Perhaps the authors will need a more specific analysis to determine if the demographic expansion is a recent event of an old event. Raggedness index is the most straightforward way to answer this question, but I would suggest more refined methods such as Bayesian Skyline Plot (BSP) in BEAST/Tracer.

Ln 268-269. “In this study, the COI gene divergence between the two lineages of A. triastriatus was 6.3%, which is neither in the intraspecific or interspecific range.” What does that statement mean?

Ln 329. “(…) and the network analysis showed that hap_14 was ancestral to all other haplotypes of lineage A.” As the network is unrooted, one cannot infer ancestry relationships solely based on this analysis. It is often the case people interpret the haplotype frequency and centrality as evidence of ancestry, but sometimes that is not the case.

Author Response

Minor corrections and suggestions

The maps presented are difficult to read. Perhaps the low resolution or the small font size makes it very difficult to read its content. Border and margins are also too thin. Figure 2, for instance, is very informative and the colored maps make a very compelling argument for the allopatric differentiation of linages and sub-lineage; however, the maps have insufficient resolution and are mostly faded. I would suggest that those maps be replotted with a thicker margin and better resolution.

Response: Thank you very much for your suggestion, we have revised Figure 1, Figure 2 and Figure 7 with thicker margin and better resolution in the revised Manuscript.

Ln 165-166. “The haplotype network was constructed using a 679 bp fragment of COI from 65 individuals of 35 populations.” I suggest changing the word “populations” for a more adequate term such as “collection sites” or “locations.” One of the main objectives of this work is to determine the number of gene pools (i.e., real populations), which the authors later found to be two and not 35.

Response: Yes, we agree with your opinion, we have changed “populations” to “locations” in Ln 13, 80, 82, 94, 186 and 220 of the revised Manuscript.

Ln 174-175. “All 65 individuals of 35 population yielded high-quality (…).” Same as above.

Response: We have revised this genetic differentiation exist Ln 194.

Ln 212-213. “The value of pairwise comparisons of φST among lineages was significantly high, suggesting a lack of gene flow between lineages A and B.” As mitochondrial markers do not recombine and therefore do not allow us to conclude anything about lineage hybridization of introgression, the authors should be cautious using expressions such as the above. The data indicate strong geographical isolation and genetic structure, which means that drift is more important than gene flow. However, are the lineages isolated because of the parthenogenetic reproduction (i.e., the gene pool is inaccessible to one another)? Can they hybridize at some point in the geographical range?  

Response: Base on your advice and our further understanding of the gene flow, we have revised “gene flow” as “a great genetic differentiation exist” in Ln 209, 240, 302, 347 and 465 of the revised manuscript. In the case of A. triastriatus, lineage A with a thin and lustreless seminal chamber, no prostate gland observed, and it was almost degenerated to parthenogenesis, which means it could not reproduce sexually. Thus, the two lineages are complete reproductive isolation, there is no hybridize between them.

The statements in Figure 5.1 and 6.1 captions (“observed frequency of pairwise difference under the sudden lineage expansion model”) contradicts the statement made in the line 226-227 (“(…) were not steady-state lineages, suggesting an old expansion”). Perhaps the authors will need a more specific analysis to determine if the demographic expansion is a recent event of an old event. Raggedness index is the most straightforward way to answer this question, but I would suggest more refined methods such as Bayesian Skyline Plot (BSP) in BEAST/Tracer.

Response: Thank you very much for your suggestion. There is our first time to use Bayesian Skyline Plot (BSP) to perform some analysis about earthworm. As you suggested, we have done more researches and ask some specialists in this area to help us with our analysis. The result shown that BSP could help us to confirm that the demographic expansion of A. triastriatus is a recent event in an old expansion. We have added BSP analysis in our manuscript, result was shown in Ln 253-256, and Figure 6 of the revised Manuscript. Thank you again for this suggestion, it is not only help us to improve this manuscript but also provide us a new idea to use BSP to conduct further population structure analysis of earthworm.

Ln 268-269. “In this study, the COI gene divergence between the two lineages of A. triastriatus was 6.3%, which is neither in the intraspecific or interspecific range.” What does that statement mean?

Response: When it comes to taxonomy of earthworm, we usually use the COI gene divergence values combined with morphological descriptions to illustrate species or subspecies. In taxonomy content, previous studies shown that levels of interspecific COI barcode divergences ranging from ~14% upwards - such as the rang of interspecific COI gene divergence values carried by 321 species of the family Megascolecidae of China is 14-24%, interspecific COI gene divergence values of invasive Asian earthworms in the northeast United States ranged from 15.84% to 24.03%; the intraspecific COI divergences ranging from ~3% backwards – such as the rang of intraspecific COI gene divergence values carried by 321 species of the family Megascolecidae of China is 0-3.0%, intraspecific COI gene divergence values of Amynthas species in North America is 0.-0.4%.

Normally, if the value is larger than 14%, it should be considered as interspecific difference, the individuals should be illustrated as species; if the value is smaller than 3%, it should be considered as intraspecific difference; and if the value is neither larger than 14% nor smaller than 3%, combined with different morphological characters, the individuals should be illustrated as subspecies.

Thus, in this study, the COI gene divergence value between the two lineages of A. triastriatus is 6.3%, which is larger than 3% and smaller than 14%”, the value is neither in the intraspecific or interspecific range. Thus, combined with different morphological characters, it is proposed to erect a new subspecies of A. triastriatus.

Ln 329. “(…) and the network analysis showed that hap_14 was ancestral to all other haplotypes of lineage A.” As the network is unrooted, one cannot infer ancestry relationships solely based on this analysis. It is often the case people interpret the haplotype frequency and centrality as evidence of ancestry, but sometimes that is not the case.

Response: Yes, it is a wrong statement. Base on your advice, we revised this sentence as “the network analysis showed that hap_14 was the hap that connect to lineage B.” in Ln 382 of the revised Manuscript.

Reviewer 2 Report

The manuscript entitled "Population genetic structure reveals two lineages of Amynthas triastriatus (Oligochaeta: Megascolecidae) in China, with notes on a new subspecies of Amynthas triastriatus" is interesting.  The subject of the manuscript is consistent with the scope of the Journal. The authors applied correct analytical methods and received many interesting results. Usually the obtained results do not raise any substantive or scientific objections, are correctly interpreted and developed. The quoted literature is sufficient.

The manuscript needs improvement.

Authors should be more highlight, what is the prime novelty of the manuscript? The results of this study must be more compared with other studies. Authors should discuss the results and how they can be interpreted in perspective of previous studies and of the working hypotheses. This will improve the quality of publications. Make sure all items are cited in the text and in the table of contents and vice versa. The "Reference" section must be corrected. It is written in the wrong font now.

Author Response

The manuscript needs improvement.

Authors should be more highlight, what is the prime novelty of the manuscript? The results of this study must be more compared with other studies. Authors should discuss the results and how they can be interpreted in perspective of previous studies and of the working hypotheses. This will improve the quality of publications. Make sure all items are cited in the text and in the table of contents and vice versa. The "Reference" section must be corrected. It is written in the wrong font now.

Response: Thank you very much for your suggestions. In order to clear the classification status of A. triastriatus, we carried out the population genetic structure studies and phylogenetic studies to analysis the differentiation and diffusion of A. triastriatus, at the meanwhile, to explore the dispersal and its impactor factors during the process. We have modified some sections in “Abstract” and “Introduction” part, Ln 14-15, 17-18, 44-47, 77-78, and 83 of the revised Manuscript as your advice. We have also added some more comparisons with other studies and discuss the results and how they can be interpreted in perspective of previous studies and of the working hypotheses in “Discussion” and “Conclusion” part, Ln 292-297, 431-440, 449-452 and 472-473 of the revised Manuscript based on your suggestion. We have checked all citations in the text and in the table of contents. We are sorry to write “Reference” in the wrong front, we have revised them in the correct front according to the guide of “introductions for Authors”
